# Improving Data Communication of Enhanced Loran Systems Using 128-*ary* Polar Codes

**DOI:** 10.3390/s25154638

**Published:** 2025-07-26

**Authors:** Ruochen Jia, Yunxiao Li, Daiming Qu

**Affiliations:** School of Electronic Information and Communications, Huazhong University of Science and Technology, Wuhan 430074, China; jiaruochen@hust.edu.cn (R.J.); liyunxiao@hust.edu.cn (Y.L.)

**Keywords:** enhanced Loran (eLoran) system, polar codes, Pulse Position Modulation (PPM), non-binary polar codes

## Abstract

The enhanced Loran (eLoran) system, a critical terrestrial backup for the Global Satellite Navigation System (GNSS), traditionally utilizes a Reed-Solomon (RS) code for its data communication, which presents limitations in error performance, particularly due to its decoding method. This paper introduces a significant advancement by proposing the replacement of the conventional RS code with a 128-*ary* polar code, which is designed to maintain compatibility with the established 128-*ary* Pulse Position Modulation (PPM) scheme integral to eLoran’s positioning function. A Soft–Soft (SS) demodulation method, based on a correlation receiver, is developed to provide the requisite soft information for the effective Successive Cancellation List (SCL) decoding of the 128-*ary* polar code. Comprehensive simulations demonstrate that the proposed 128-*ary* polar code with SS demodulation achieves a substantial error performance improvement, yielding an approximate 9.3 dB gain at the 0.01 FER level over the RS code in eLoran data communication with EPD-MD demodulation. Additionally, the proposed scheme improves data transmission efficiency—either reducing transmission duration by 2/3 or increasing message bit number by 250% for comparable error performance—without impacting the system’s primary positioning capabilities.

## 1. Introduction

The enhanced Loran (eLoran) system has become a vital terrestrial backup for the Global Satellite Navigation System (GNSS), owing to its strong anti-jamming capabilities and improved operational stability [1,2,3,4,5]. A core function of eLoran is data communication, which traditionally consists of two main components: (1) 128-*ary* Pulse Position Modulation (PPM), which is fundamental to the system’s positioning capability, and (2) Reed–Solomon (RS) coding for data encoding [6]. Current research efforts to improve the error performance of eLoran data communication have primarily concentrated on refining modulation and demodulation techniques [7,8,9,10]. However, such improvements in demodulation methods typically yield only incremental gains in error performance. On the other hand, any modification to the modulation scheme risks undermining the established positioning functionality, which is critical for eLoran’s operational integrity. Therefore, enhancing the data encoding process emerges as a more promising and less disruptive approach to achieving significant improvements in error performance while maintaining full compatibility with the existing positioning infrastructure.

The RS code, introduced by Reed and Solomon in 1960 [11], is the conventional choice for eLoran data encoding. In practice, RS codes are predominantly decoded using hard-decision decoding (HDD) algorithms, such as the Berlekamp–Massey (BM) algorithm [12], the Euclidean algorithm [13], and the Welch–Berlekamp algorithm [14], with the BM algorithm being the most widely adopted for eLoran [6]. Theoretically, soft-decision decoding (SDD) offers superior error correction compared to HDD, with Maximum Likelihood (ML) decoding being optimal. But the ML decoding process for RS codes is known to be NP-hard [15], making it impractical for practical applications. And existing SDD algorithms for RS codes, such as the chase Kötter-Vardy algorithm [16], partially parallel low-complexity chase algorithm [17], and the Bit-level Generalized Minimum Distance (BGMD) algorithm [18], provide only limited performance gains at the cost of significantly increased complexity. This situation highlights the inherent limitations of RS codes in further improving the error performance of eLoran data communication.

Given that eLoran data are transmitted in short-code, this paper proposes the adoption of polar codes, which are particularly well-suited for short-code applications. Polar codes, proposed by Arikan [19], are theoretically proven to achieve the capacity of arbitrary symmetric discrete memoryless channels (DMCs) and were adopted in the 5G mobile communication standard in 2016 [20]. The fundamental principle underlying polar codes is channel polarization, which transforms a set of identical binary-input DMCs into a series of polarized subchannels with widely varying channel error rates (CERs): some subchannels exhibit CERs approaching zero, while others approach one. Information is transmitted over the most reliable subchannels—those with CERs near zero. The classic SDD algorithm for polar codes is the Successive Cancellation (SC) decoder, also proposed by Arikan [19], which has a complexity of $O(N\cdot {log}_{2}N)$, where N is the code length.

In this work, we propose the adoption of a 128-*ary* polar code specifically designed for the eLoran system, aiming to replace the conventional RS code while preserving the 128-*ary* PPM. This ensures that the proposed enhancement does not compromise the system’s core positioning function. Since the SC decoder for polar codes requires soft information as input, we introduce a Soft-Soft (SS) demodulation method based on a correlation receiver. The main contributions of this paper are as follows:The design of a 128-*ary* polar code that is compatible with the 128-*ary* PPM employed in the eLoran system;The proposal of an SS demodulation method, based on a correlation receiver, designed for 128-*ary* PPM to provide the requisite soft information for the 128-*ary* polar decoder;A demonstration that the proposed 128-*ary* polar code with SS demodulation achieves an error performance gain of approximately 9.3 dB compared to conventional eLoran data encoding (RS codes with conventional demodulation). The origins of this significant gain are analyzed in the subsequent sections.

The remainder of this paper is organized as follows. Section 2.1 details the conventional eLoran data encoding and modulation methods, while Section 2.2 presents the design of the 128-ary polar code for the eLoran system. Section 3 provides the simulation results comparing error performance. Finally, Section 4 discusses the findings and concludes this paper.

## 2. Materials and Methods

This section presents the fundamental techniques and system parameters underpinning the eLoran data communication architectures evaluated in this study. We first provide a detailed exposition of the conventional eLoran data communication system, which employs RS coding and 128-*ary* PPM. Subsequently, we introduce the proposed 128-*ary* polar coding scheme, specifically designed to enhance the error performance of eLoran data transmission while maintaining full compatibility with the established modulation and positioning framework. Figure 1. offers a comparative overview of both system models: Figure 1a illustrates the traditional eLoran data communication system, while Figure 1b depicts the proposed system incorporating the 128-*ary* polar code. In both configurations, the original $K^{\prime \prime }$ message bits are first processed by a Cyclic Redundancy Check (CRC) encoding, yielding $K^{\prime }$ bits (the original message bits with appended CRC parity). These $K^{\prime }$ bits are then mapped to K 128-*ary* symbols. These symbols are subsequently encoded using the RS code or the 128-*ary* polar code. Following 128-*ary* PPM, the modulated pulses are transmitted over an Additive White Gaussian Noise (AWGN) channel. At the receiver, the signals are demodulated, and the resultant sequences are processed by the corresponding RS or 128-*ary* polar decoder. Then, a final CRC verifies the integrity of the recovered message.

The primary distinction between the systems, as illustrated by the comparison in Figure 1a,b, lies in the encoding/decoding strategies, along with a corresponding refinement of the demodulation process for the 128-*ary* polar code scheme. Importantly, the 128-*ary* PPM scheme remains unchanged. This consistency ensures that improvements in data encoding do not compromise the established positioning functionality of the eLoran system, which is a key objective of this work.

### 2.1. Conventional eLoran Data Communication System

The conventional data encoding process for the eLoran system, specified by Eurofix [6] and consistent with ITU M589-3 recommendations, utilizes a CRC-aided RS code defined over the Galois Field GF(q) where q=128. As illustrated in Figure 1a, the encoding process begins with $K^{\prime \prime }=56$ message bits, which is first encoded by a 14-bit CRC encoder, characterized by the following generator polynomial:(1)$$G_{CRC}\left(x\right)=x^{14}+x^{13}+x^{7}+x^{5}+x^{4}+1,$$
resulting in a $K^{\prime }=70$ bit sequence. These 70 bits are then mapped into K=10 information symbols, each represented as a 7-bit element of GF(128). These K symbols are subsequently encoded by an RS(M,K) code, where M=30 is the code length. The field GF(128) is constructed using the primitive polynomial f(x):(2)$$f(x)=x^{7}+x^{3}+1,$$
and the generator polynomial $G_{RS}(x)$ for the RS code is given by the following:(3)$$G_{RS}\left(x\right)=\left(x-a\right)\cdot \left(x-a^{2}\right)\cdot \left(x-a^{3}\right)\cdot \ldots \cdot \left(x-a^{20}\right).$$
where a is a primitive element of GF(128) derived from *f*(*x*).

For data modulation, the traditional eLoran system employs 128-*ary* PPM, as detailed in Eurofix [6]. Each of the M=30 RS-coded symbols is transmitted within a Group Repetition Interval (GRI). Each GRI consists of 6 standard eLoran pulses, and each pulse can be modulated to one of three time-slot positions (see Figure 2): 1 μs advance, prompt (no shift), or 1 μs delay. This three-level modulation across 6 pulses yields $3^{6}$ = 729 possible pulse sequences per GRI.

From the 729 possible sequences, 141 are designated as “balanced” sequences (where the counts of advanced, prompt, and delayed pulses within a GRI are equal, i.e., two of each type). According to Eurofix [6], from this set of 141 balanced sequences, 128 are selected to represent the 128 distinct symbol values $j\in \left\{0,\;1,\ldots ,q-1\right\}$, where q=128. These selected sequences are called “standard modulation sequences” in this paper. The modulation level of the h-th pulse $\left(h\in \left\{1,\;2,\ldots ,6\right\}\right)$ in a standard modulation sequence corresponding to the symbol value j is denoted as $S(h,j)\in \left\{0,1,2\right\}$, where $\left\{0,1,2\right\}$ corresponds to 1 μs advance, prompt, and 1 μs delay, respectively.

At the receiver, the conventional eLoran system typically uses the Envelope Phase Detection–Majority Decision (EPD–MD) demodulation technique [8]. The output of the EPD-MD demodulator provides symbol estimates to the RS decoder, and the integrity of the final recovered message is verified via a CRC.

### 2.2. The Proposed 128-ary Polar Coding Scheme for eLoran

To enhance the error performance of the eLoran data communication system, this section introduces an encoding strategy based on 128-*ary* polar codes. This approach is specifically designed to operate in conjunction with the existing 128-*ary* PPM scheme, thereby maintaining compatibility with the system’s established positioning functions. At the receiver, a corresponding Soft–Soft (SS) demodulation technique is developed to provide the necessary soft information for the decoder employed for the 128-*ary* polar codes.

It is important to note that the primary aim of this investigation is to demonstrate the viability and potential benefits of substituting the conventional RS code with the 128-*ary* polar code in the eLoran context. As such, the specific 128-*ary* polar code encoding/decoding algorithms, construction method, and rate-matching techniques presented here serve as an initial framework and are not necessarily optimized and further refinements may yield additional performance gains.

#### 2.2.1. Polar Codes for 128-*ary* PPM 

When adapting polar codes for a 128-*ary* PPM system, two principal architectural choices are possible: (1) a binary polar code with the bit-interleaved coded modulation (BICM) [21] scheme; and (2) a 128-*ary* polar code with 128-*ary* modulation scheme (non-binary q-*ary* polar codes have been discussed in [22,23,24]). Directly employing a 128-*ary* polar code with 128-*ary* modulation can potentially offer superior error performance compared to a binary polar code with BICM. While this non-binary approach typically entails higher computational complexity, the inherent low data rate of the eLoran system mitigates this concern, rendering the complexity of a 128-ary polar code acceptable for this application. Consequently, this work primarily focuses on the design and evaluation of a 128-*ary* polar code. For comparative analysis, a binary polar code with BICM is also implemented and assessed in Section 3.

The input to the proposed polar coding scheme, as depicted in Figure 1b, originates from $K^{\prime \prime }=56$ message bits. These are first encoded using a 14-bit CRC encoder, yielding a $K^{\prime }=70$ bit sequence (comprising the original message and CRC parity bits). This 70-bit sequence is then converted into K=10 information symbols, each belonging to GF(128). The 128-*ary* polar encoder operates on these K symbols. In this work, we consider a q-*ary* polar code with q=128. The mother code length is chosen as N=32, which is the power of 2 ($N=2^{n}$, where n=5) closest to the target transmitted block length of M=30 symbols. The input vector to the polar encoder, $u_{1}^{N}=(u_{1},u_{2},u_{3},\ldots ,u_{N})$, consists of K=10 information symbols and N−K frozen symbols. These symbols are placed on the K most reliable subchannels, which are indexed by the information set A(A⊆{1,2,…,N}), with $\left|A\right|=K$. The method of obtaining A will be explained in Section 2.2.3. The remaining N−K subchannels are indexed by the frozen set $A^{c}$ carrying frozen symbols, which are set to zero. The polar codeword $c_{1}^{N}=(c_{1},c_{2},c_{3},\ldots ,c_{N})$ is given by the following:(4)$$c_{1}^{N}=\mathrm{mod}(u_{1}^{N}\cdot F^{⊗n},q)\cdot B_{N},$$
where $B_{N}$ is a bit-reversal permutation matrix and $F^{⊗n}$ is the n-th Kronecker power of F over GF(q), where F is the kernel. In this paper, a 2×2 non-binary kernel based on [24] is chosen, which is described as(5)$$F=\left[\begin{array}{cc} 1 & 0\\ \alpha & \beta \end{array}\right],\;\;\;\;\alpha ,\beta \in GF(q),$$

For this implementation, as introduced in [24], we select α=57 and β=1, with q=128.

Since the length of the mother codeword $c_{1}^{N}$ (N=32) differs from the desired transmission length (M=30), rate matching is achieved via puncturing. In this work, the Quasi-Uniform Puncturing (QUP) algorithm [25] is employed. The QUP process follows the bit-reversal order. By applying bit-reversal to the codeword indices $\left[1,2,3,\ldots ,N\right]$, we obtain the sequence [1,17,9,25,…,16,32]. In this work, since two coded symbols need to be punctured, the coded symbols with the first two indices in this sequence (index 1 and index 17) from the mother codeword $c_{1}^{N}$ are selected for puncturing. This results in a shortened codeword ${\tilde{c}}_{1}^{M}$ of length M=30 symbols. Each symbol in ${\tilde{c}}_{1}^{M}$ is then modulated using the 128-*ary* PPM scheme detailed in Section 2.1, resulting in 30 GRIs for transmission.

At the receiver, the demodulation process (detailed in Section 2.2.2) for the received sample sequences yields a demodulation output ${\tilde{y}}_{1}^{M}=({\tilde{y}}_{1},{\tilde{y}}_{2},\ldots ,{\tilde{y}}_{M})$ of M symbols. Each element ${\tilde{y}}_{m}(m\in \left\{1,\;2,\ldots ,M\right\})$ is a q-dimensional vector, and ${\tilde{y}}_{m}=({\tilde{y}}_{m}\left(0\right),{\tilde{y}}_{m}\left(1\right),\ldots ,{\tilde{y}}_{m}\left(q-1\right))$, where ${\tilde{y}}_{m}(j)$ for j∈{0,1,…,q−1} represents the demodulation output with the possible symbol value j for the m-th received symbol.

Before decoding, this sequence ${\tilde{y}}_{1}^{M}$ is de-punctured to reconstruct a sequence $y_{1}^{N}=(y_{1},y_{2},\ldots y_{N})$ of length N, corresponding to the mother code. Each element $y_{i}(i\in \left\{1,\;2,\ldots ,N\right\})$ is a q-dimensional vector, $y_{i}=(y_{i}\left(0\right),y_{i}\left(1\right),\ldots ,y_{i}\left(q-1\right))$. This involves inserting erasure information at the punctured positions (1st and 17th). For these positions, the corresponding vectors $y_{i}=\emptyset (i\in \left\{1,17\right\})$ for all q possible symbol values, indicating that these symbols were not transmitted. For the non-punctured positions, $y_{i}$ = ${\tilde{y}}_{k_{i}}$($i\in \{1,2,\ldots ,N\}\backslash \left\{1,17\right\})$, where $k_{i}\in \{1,2,\ldots ,M\}$ maps to the i-th non-punctured index.

For the decoder, let $W_{N}^{\left(i\right)}(y_{1}^{N},{\hat{u}}_{1}^{i-1}|u_{i})(i\in \left\{1,\;2,\ldots ,N\right\})$ denote the transition probability of $u_{i}$ under $y_{1}^{N}$ and the previously estimated symbols ${\hat{u}}_{1}^{i-1}=({\hat{u}}_{1},{\hat{u}}_{2},\ldots ,{\hat{u}}_{i-1})$ of $u_{1}^{i-1}$. Let $L_{N,j}^{\left(i\right)}(y_{1}^{N},{\hat{u}}_{1}^{i-1})$ denote the Log-Likelihood Ratio (LLR) of $u_{i}$ with symbol value j∈{0,1,…,q−1} under the demodulation outputs $y_{1}^{N}$ and the previously estimated symbols ${\hat{u}}_{1}^{i-1}$. The $L_{N,j}^{\left(i\right)}(y_{1}^{N},{\hat{u}}_{1}^{i-1})$ is computed as follows:(6)$$L_{N,j}^{\left(i\right)}(y_{1}^{N},{\hat{u}}_{1}^{i-1})=\ln\frac{W_{N}^{\left(i\right)}(y_{1}^{N},{\hat{u}}_{1}^{i-1}|u_{i}=0)}{W_{N}^{\left(i\right)}(y_{1}^{N},{\hat{u}}_{1}^{i-1}|u_{i}=j)},$$

The initial value of the decoder is the LLRs of $y_{i}$ with symbol value j, denote as $L_{1,j}^{\left(1\right)}(y_{i})=\ln\frac{W(y_{i}|0)}{W(y_{i}|j)}$, where $L_{1,j}^{\left(1\right)}\left(y_{i}\right)=0$ if $y_{i}=\emptyset$. The estimated symbol ${\hat{u}}_{i}$ is generated as follows:(7)$${\hat{u}}_{i}=\left\{\begin{array}{cc} \underset{j}{\mathrm{argmin}}\left(L_{N,j}^{\left(i\right)}(y_{1}^{N},{\hat{u}}_{1}^{i-1})\right), & if\;\;\underset{j}{\min}\left(L_{N,j}^{\left(i\right)}(y_{1}^{N},{\hat{u}}_{1}^{i-1})\right)<0\\ 0, & otherwise\; \end{array}\right.,$$

For the decoding method, the SC decoding method for q-*ary* polar codes described in [23] is referred and extended to work with a 2×2 non-binary kernel:(8)$$\begin{array}{ll} L_{N,j}^{\left(2i-1\right)}\left(y_{1}^{N},{\hat{u}}_{1}^{2i-2}\right) & =\ln\frac{\sum e^{-(L_{N/2,0}^{\left(i\right)}(y_{1}^{N/2},X)+L_{N/2,r}^{\left(i\right)}(y_{N/2+1}^{N},X^{\prime }))}}{\sum e^{-(L_{N/2,j}^{\left(i\right)}(Y_{1}^{N/2},X)+L_{N/2,r}^{\left(i\right)}(Y_{N/2+1}^{N},X^{\prime }))}}\\ & \approx \underset{r\in \left[0,1,\ldots ,q-1\right]}{\max}\left(-(L_{N/2,0}^{\left(i\right)}(y_{1}^{N/2},X)+L_{N/2,r}^{\left(i\right)}(y_{N/2+1}^{N},X^{\prime }))\right)\\ & -\underset{r\in \left[0,1,\ldots ,q-1\right]}{\max}\left(-(L_{N/2,j}^{\left(i\right)}(y_{1}^{N/2},X)+L_{N/2,r}^{\left(i\right)}(y_{N/2+1}^{N},X^{\prime }))\right), \end{array}$$
and(9)$$\begin{array}{cc} L_{N,j}^{\left(2i\right)}\left(y_{1}^{N},{\hat{u}}_{1}^{2i-1}\right) & =(-(L_{N/2,{\hat{u}}_{2i-1}}^{\left(i\right)}(y_{1}^{N/2},X)+L_{N/2,0}^{\left(i\right)}(y_{N/2+1}^{N},X^{\prime })))\\ & -(-(L_{N/2,{\hat{u}}_{2i-1}}^{\left(i\right)}(y_{1}^{N/2},X)+L_{N/2,j}^{\left(i\right)}(y_{N/2+1}^{N},X^{\prime }))), \end{array}$$
where X and $X^{\prime }$ are defined as follows [19]:(10)$$X=\begin{array}{cc} 1\cdot {\hat{u}}_{1,o}^{2i-2}+\alpha \cdot {\hat{u}}_{1,e}^{2i-2} & ,on\;GF(q) \end{array},\;$$(11)$$X^{\prime }=\begin{array}{cc} 0\cdot {\hat{u}}_{1,o}^{2i-2}+\beta \cdot {\hat{u}}_{1,e}^{2i-2} & ,on\;GF\left(q\right) \end{array}.\;$$
where ${\hat{u}}_{1,o}^{2i-2}$ and ${\hat{u}}_{1,e}^{2i-2}$ are sub-vectors of ${\hat{u}}_{1}^{2i-2}$ with odd and even indices.

Equations (8) and (9) define the recursive structure of the Successive Cancellation (SC) decoder. As SC is a soft-decision decoding (SDD) algorithm, it requires soft inputs from the demodulator. To provide these inputs, we developed an SS demodulation method, detailed in Section 2.2.2.

A well-known limitation of the standard SC decoder is its susceptibility to error propagation, where an incorrect decision at an early stage can corrupt all subsequent decoding steps. To mitigate this issue, we employ the Successive Cancellation List (SCL) decoder, an enhanced algorithm proposed in [26], which approaches Maximum Likelihood (ML) performance. The SCL decoder mitigates error propagation by maintaining a list of L most likely candidate paths throughout the decoding process. Following SCL decoding, the decoding results that satisfy the CRC are selected as the final recovered message.

#### 2.2.2. Proposed Demodulation Method

Conventional demodulation methods for the eLoran system, such as Signal Matching Correlation–Pulse Position Detection (SMC–PPD) [7] and Envelope Phase Detection–Majority Decision (EPD–MD) [8], were designed with significant constraints on computational complexity, rendering them suboptimal in terms of error performance. Given the substantial advancements in modern processing capabilities, this work employs an optimal demodulator, the correlation receiver, to achieve optimal error performance. To this end, we model the demodulation process in two hierarchical layers: (1) The first layer consists of the demodulation of each pulse. (2) The second layer consists of the demodulation of each GRI.

Based on this two-layer framework, we propose a Soft–Soft (SS, soft demodulation with two layers) demodulation method based on correlation receiver to work with the SCL decoder of the 128-*ary* polar code. Additionally, to demonstrate the source of the error performance gains in the proposed method, we also design Soft–Hard (SH, the first layer is soft demodulation and the second layer is hard demodulation) and Hard–Hard (HH, hard demodulation in two layers) demodulation methods to work with the RS code.

In the first layer of the SS demodulation, the received sample sequence of the h-th $\left(h\in \left\{1,\;2,\ldots ,6\right\}\right)$ pulse in the m-th $\left(m\in \left\{1,\;2,\ldots ,M\right\}\right)$ received GRI is denoted as $R_{m,h}=\left(r_{m,h}^{1},r_{m,h}^{2},\ldots ,r_{m,h}^{T}\right)$. And a modulated standard eLoran pulse with modulation level $s\in \left\{0,1,2\right\}$(corresponding to 1 μs advance, prompt and 1 μs delay) is denoted as $P_{s}=(p_{s}^{1},p_{s}^{2},\ldots ,p_{s}^{T},)$, where T is the number of samples per pulse. Let ${\tilde{y}}_{m,h}^{s}$ denote the Euclidean Distance of $R_{m,h}$ and $P_{s}$. Therefore, under ideal synchronization conditions, ${\tilde{y}}_{m,h}^{s}$ is calculated as follows:(12)$${\tilde{y}}_{m,h}^{s}=\sum_{t=1}^{T}{\left(r_{m,h}^{t}-p_{s}^{t}\right)}^{2},$$
where t represents the index of each sample in $R_{m,h}$ and $P_{s}$.

Then, in the second layer of SS demodulation, let ${\tilde{y}}_{m}(j)$ denote the Euclidean Distance of the m-th received GRI with the modulation sequence corresponding to symbol value $j\in \left\{0,\;1,\ldots ,q-1\right\}$:(13)$${\tilde{y}}_{m}(j)=\sum_{h=1}^{H}{\tilde{y}}_{m,h}^{S(h,j)},$$
where H=6 is the total number of pulses in a received GRI and $S(h,j)\in \left\{0,1,2\right\}$ (corresponding to 1 μs advance, prompt and 1 μs delay) denote the modulation level of the h-th pulse $\left(h\in \left\{1,\;2,\ldots ,6\right\}\right)$ in a standard modulation sequence corresponding the symbol value $j\in \left\{0,\;1,\ldots ,q-1\right\}$. The result of the second layer of soft demodulation is denoted as ${\tilde{y}}_{1}^{M}=({\tilde{y}}_{1},{\tilde{y}}_{2},\ldots ,{\tilde{y}}_{M})$, in which ${\tilde{y}}_{m}=[{\tilde{y}}_{m}\left(0\right),{\tilde{y}}_{m}\left(1\right),\ldots ,{\tilde{y}}_{m}(q-1)]$.

After SS demodulation, as introduced in Section 2.2.1, ${\tilde{y}}_{1}^{M}$ is de-punctured to reconstruct $y_{1}^{N}=(y_{1},y_{2},\ldots y_{N})(N=32)$, in which $y_{i}=\emptyset$ for $i\in \left\{1,17\right\}$, and $y_{i}$ = ${\tilde{y}}_{k_{i}}$($i\in \{1,2,\ldots ,N\}\backslash \left\{1,17\right\})$, where $k_{i}\in \{1,2,\ldots ,M\}$ maps to the i-th non-punctured index. Then, let $W(y_{i}|j)(i\in \left\{1,\;2,\ldots ,N\right\})$ denote the transition probability of $y_{i}$ with the condition that symbol $j\in \left\{0,\;1,\ldots ,q-1\right\}$ passesthrough an Additive White Gaussian Noise (AWGN) channel. $W(y_{i}|j)$ is obtained as follows:(14)$$W(y_{i}|j)=(\frac{1}{\sqrt{2\pi }\sigma }{)}^{T\cdot H}\cdot e^{\frac{y_{i}(j)}{2\sigma^{2}}},$$
where $\sigma^{2}$ is the noise variance in the AWGN channel. Then, LLR $L_{1,j}^{\left(1\right)}(y_{i})$ is defined as follows:(15)$$L_{1,j}^{\left(1\right)}(y_{i})=\ln\frac{W(y_{i}|0)}{W(y_{i}|j)}=\frac{y_{i}(j)}{2\sigma^{2}}-\frac{y_{i}(0)}{2\sigma^{2}},$$
where $L_{1,j}^{\left(1\right)}(y_{i})$ is the input of the SCL decoder, with $L_{1,j}^{\left(1\right)}\left(y_{i}\right)=0$ if $y_{i}=\emptyset$.

The Soft-Hard (SH) demodulation method is the simplified version of the SS demodulation method. And in Section 3, we use SH demodulation to work with the RS decoder to make a comparison with the proposed scheme and demonstrate the source of the performance gain. In the SH demodulation method, the first layer of soft demodulation is the same as that in the SS demodulation method. However, the second layer of SH demodulation for each symbol is given as hard decision z(m):(16)$$z\left(m\right)=\underset{j\in [0,1,\ldots ,q-1]}{\mathrm{argmin}}\left({\tilde{y}}_{m}(j)\right),$$
which is then used as the input for the RS decoder.

The Hard–Hard (HH) demodulation method is a further simplified version of the SS demodulation method. HH demodulation is also used to work with the RS decoder to make a comparison in Section 3 with the proposed scheme. In the first layer of hard demodulation, let ${\hat{s}}_{m,h}$ denote the decided modulation level of h-th $\left(h\in \left\{1,\;2,\ldots ,6\right\}\right)$ pulse. And ${\hat{s}}_{m,h}$ is calculated as follows:(17)$${\hat{s}}_{m,h}=\underset{s\in [0,1,2]}{\mathrm{argmin}}\left({\tilde{y}}_{m,h}^{s}\right),$$

Then, in the second layer of the hard demodulation of HH, the hard decision $z\left(m\right)$ is obtained as follows:(18)$$z\left(m\right)=\underset{j\in [0,1,\ldots ,q-1]}{\mathrm{argmin}}(\sum_{h=1}^{H}{\left({\hat{s}}_{m,h}-S(h,j)\right)}^{2}).$$
which is the input of the RS decoder.

In comparison to the HH demodulation method, the traditional eLoran system employs methods with even lower computational complexity, such as the EPD-MD method [8]. And in Section 3, we provide a comparison of the error performance of the 128-*ary* polar code with SS demodulation and the RS code with SH demodulation/HH demodulation/EPD-MD demodulation to explain the source of performance gain.

#### 2.2.3. Construction of 128-ary Polar Code for eLoran

The performance of a polar code is critically dependent on the selection of the information set A as defined in Section 2.2.1, which comprises the indices of the subchannels designated to carry information symbols. The process of identifying A is known as polar code construction, and commonly used methods for polar code construction include the Bhattacharyya parameter [19], density evolution [27], Gaussian approximation [28], Monte Carlo simulation [29], and the polarization weight method [30].

In this paper, we adopt the Monte Carlo construction method to incorporate the specific characteristics of 128-*ary* PPM and the proposed SS demodulation into the construction process. In the Monte Carlo construction method, a large number of trials are simulated for each polarized subchannel to accurately estimate its channel error rate (CER). Crucially, these simulations incorporate both 128-ary PPM and the proposed SS demodulation to yield realistic CERs for the system. Subsequently, the CERs of all polarized subchannels are sorted. The K subchannels with the smallest CERs are selected for transmitting the information symbols, forming the information set A. The indices of the remaining subchannels constitute the frozen set, $A^{c}$.

For each trial of the Monte Carlo construction, we first perform 128-*ary* polar encoding on a random sequence with length N=32 to generate a codeword. Subsequently, the QUP algorithm is applied to produce a punctured codeword of length M=30. This punctured codeword is then mapped to 128-*ary* PPM and transmitted over an Additive White Gaussian Noise (AWGN) channel. The channel operates at a signal-to-noise ratio (SNR) of 2.5 dB, with the specific definition of the SNR provided in Section 3. At this SNR, the largest CER among the K smallest CERs is approximately 0.01. Following ideal synchronization, the received signal undergoes SS demodulation, which is then followed by SC decoding.

By repeatedly executing this process $10^{6}$ times, the number of errors for the i-th $\left(i\in \left\{1,\;2,\ldots ,N\right\}\right)$ polarized subchannel is counted. During this error counting, it is assumed that the decoding results of the preceding subchannels (from the 1st to the (i−1)th) are perfectly known and correct. Subsequently, the CERs are sorted, and the indices of the K=10 polarized subchannels exhibiting the lowest CERs are designated as the information set A.

Figure 3 illustrates the resulting CERs. As expected, for one subset of polarized subchannels, the CER approaches zero, while for another subset, the CER approaches one (specifically, 127/128). As indicated by the ‘*’ markers in Figure 3, the K=10 subchannels with the smallest CERs are selected to form the information set A={16,23,24,26,27,28,29,30,31,32}, and the ‘·’ markers represent the frozen set $A^{c}$. Notably, the subchannels numbered 31 and 32 are reserved for CRC symbols.

## 3. Results

### 3.1. Comparative Error Performance

This section presents a comprehensive evaluation of the proposed 128-*ary* polar code, comparing its error performance against the RS code using computer simulation. The Frame Error Rate (FER) is used to evaluate the error performance under various SNRs over the AWGN channel. The simulations are conducted under ideal synchronization conditions and the number of frames used for the simulation is $10^{6}$. The parameters in the simulation are presented in Table 1, and the SNR is defined as follows:(19)$$SNR=10\cdot {log}_{10}\frac{P_{s}}{P_{n}}.$$
where $P_{s}$ is the signal power within a 20 kHz bandwidth, and $P_{n}$ is the noise power within the same bandwidth. According to the authors of [31], the signal power $P_{s}$ is calculated as $P_{s}={\left(0.506\;*\;A_{mp}\right)}^{2}$, where $A_{mp}$ is the peak amplitude of the eLoran pulse.

For a broader comparison, a binary polar code with the BICM [21] scheme for 128-*ary* PPM is also simulated. In the BICM scheme, the codeword of the binary polar code is interleaved and divided into 7-bits groups, and each 7-bits group is mapped to a 128-*ary* PPM symbol.

The simulation results, depicted in Figure 4, compare the FER performance of the proposed 128-*ary* polar code with SS demodulation against the conventional RS code paired with various demodulation techniques. Taking both complexity and error performance into account, we select the 128-*ary* polar code with L=4 as the proposed scheme. This choice is based on the observation from Figure 4 that increasing L from 2 to 4 yields a notable gain of approximately 0.4 dB. Since the calculation complexity is linearly related to the list size L (as discussed in Section 4), further increasing L to 8 doubles the calculation complexity while providing a negligible gain of approximately 0.2 dB. Thus, L=4 represents an efficient trade-off. Notably, the proposed scheme demonstrates a substantial performance advantage, achieving an SNR gain of approximately 9.3 dB at the 0.01 FER level over the conventional RS code with EPD-MD demodulation—a widely adopted baseline for eLoran receivers—at representative FERs. A detailed breakdown of this cumulative 9.3 dB gain at the 0.01 FER level, also summarized in Table 2, reveals contributions from the following different system enhancements:Replacing the EPD-MD demodulator with a correlation receiver (HH demodulation) for the RS code yields an initial gain of approximately 1.2 dB. This improvement is attributed to the optimal nature of the correlation receiver compared to the simpler EPD-MD.Further incorporating soft demodulation at the first layer (SH demodulation) with the RS code provides an additional gain of approximately 2.1 dB over the HH demodulation. This result highlights the benefit of soft information.The most significant contribution, approximately 6.0 dB, arises from replacing the RS code (with SH demodulation) with the proposed 128-*ary* polar code utilizing SCL decoding. This substantial gain underscores the superior error-correcting capability of polar codes when coupled with an effective soft-decision decoding algorithm like SCL, which inherently benefits from the soft inputs provided by the SS demodulator.

As shown in Figure 4, while the binary polar code employing the BICM scheme with SS demodulation surpasses the performance of the RS code with EPD-MD, it is still significantly outperformed by the proposed 128-*ary* polar code with SS demodulation. The BICM scheme is fundamentally suboptimal over the AWGN channel due to its separation of coding and modulation processes [32]. By computing LLRs for each bit independently, the demodulation fails to capture the joint probabilistic information present within a single modulation symbol. In contrast, the proposed 128-ary polar code adopts a non-binary framework that is intrinsically integrated with the modulation scheme. The decoder processes symbol-level likelihoods directly, thereby retaining the complete channel information. This unified design eliminates the information loss characteristic of BICM. This is the main source of the observed performance gain. This suggests that directly designing a 128-*ary* polar code for the 128-*ary* PPM, as is carried out in this work, is more effective than using a binary polar code with BICM.

Subsequent simulations further demonstrate that the proposed 128-*ary* polar coding scheme significantly enhances data communication efficiency. Figure 5 illustrates scenarios (with parameters detailed in Table 3) where the proposed system maintains an FER performance that is comparable to the baseline RS code with EPD-MD, but under significantly more favourable transmission parameters, as follows:Specifically, when maintaining the original 56-bit message length, using the 128-*ary* polar code with SS demodulation allows the number of GRIs required for transmission to be reduced by 2/3 (from 30 to 10 GRIs), while maintaining a similar FER performance.Alternatively, for a fixed transmission duration of 30 GRIs, the number of message bits can be increased by 250% (from 56 to 175 bits) while achieving a similar FER to the conventional system.These results underscore the potential of the proposed scheme to substantially increase data communication efficiency, enhancing the communication capabilities of the eLoran system.

To elucidate the critical role of soft-decision decoding (SDD) in achieving the observed gains with polar codes, an additional simulation was conducted. Figure 6 compares the 128-*ary* polar code (30 GRI and 56 bits) using SH demodulation with the RS code also using SH demodulation. In this setup, the SH demodulator provides hard decisions on symbols to the SCL decoder, effectively creating an HDD scenario for the 128-*ary* polar code. The results indicate that when constrained to HDD, the 128-*ary* polar code performs approximately 1.6 dB worse than the RS code. This finding strongly suggests that the substantial performance benefits of the 128-*ary* polar code are intrinsically linked to its effective utilization of soft information through SDD (specifically, SCL decoding).

### 3.2. Doppler Shift Error Performance Sensitivity

To evaluate the impact of relative motion on error performance, this subsection examines the Doppler shifts that can arise in the eLoran system and analyzes their effect on the eLoran communication schemes.

Since the eLoran system typically involves ground-based transmission and reception, we first consider a mobile communication scenario with a maximum relative velocity of 100 km/h, according to the Doppler shift formula $∆f=\frac{v}{c}\cdot f_{c}$, where ∆f is the Doppler shift, v is the maximum relative velocity between the source and the receiver, $c=3\times 10^{8}$ m/s is the light velocity, and $f_{c}=100$ kHz is the eLoran signal frequence [6]. The resulting frequency shift is approximately 0.0093 Hz, which is negligible. Furthermore, we consider the case of low earth orbit (LEO) satellite communication. Assuming a satellite-to-ground station distance of 550 km (corresponding to the Starlink satellites), the maximum relative velocity between the satellite and the ground station can reach approximately 7.5 km/s, resulting in a Doppler shift of about 2.5 Hz. We conducted error performance simulations under a 2.5 Hz Doppler shift, and the results (presented in Figure 7) are nearly identical to those obtained without Doppler shift, regardless of the coding and decoding methods employed. The above results indicate that the insensitivity to Doppler shift is a characteristic of eLoran modulation and is independent of the encoding and decoding methods used. This is because the principle of eLoran is primarily based on the pulse’s time of arrival, rather than carrier phase or frequency, making the impact of Doppler shift on the eLoran system negligible. Therefore, the Doppler shift error performance sensitivity will not be further discussed in the following sections.

## 4. Discussion

The simulation results presented herein compellingly demonstrate the substantial benefits of adopting the proposed 128-*ary* polar codes with SS demodulation for the eLoran data communication system. This approach offers significant advantages in error performance and data transmission efficiency compared to the conventional RS code. Specifically, our findings highlight two key advantages:1.Enhanced Error Performance: For equivalent message lengths and the number of GRIs, the 128-*ary* polar code with SS demodulation achieves an approximate 9.3 dB SNR gain over the conventional RS code with EPD-MD demodulation.2.Improved Data Communication Efficiency: Alternatively, while maintaining a comparable error performance to the traditional system, the proposed scheme can reduce the required number of GRIs by 2/3 (from 30 to 10 for a 56-bit message) or increase the message bit number by 250% (from 56 to 175 bits within 30 GRIs).

A systematic analysis of the simulations and comparisons reveals the contributing factors to this performance gain. The adoption of a correlation receiver (HH demodulation) provides an initial improvement over EPD-MD demodulation. The introduction of soft-decision demodulation (transitioning from HH to SH demodulation for the RS code) further enhances performance by preserving valuable channel information. However, the most significant gain is realized through the combination of the 128-*ary* polar code with SDD, specifically using the SCL algorithm. Our results (Figure 6) underscore this point: when the 128-*ary* polar code is constrained to hard-decision inputs (via SH demodulation), its performance degrades, even falling below that of the RS code under similar hard-decision conditions.

While one might consider applying SDD techniques to the RS code, the practical challenges are considerable. As noted in [15], the optimal Maximum Likelihood (ML) decoding of RS codes is an NP-hard problem, rendering it computationally infeasible for real-time systems. Existing sub-optimal SDD algorithms for RS codes typically offer only modest gains over HDD and often come with a high complexity penalty. In contrast, the 128-ary polar code with SCL decoding offers a practical and effective SDD solution with acceptable complexity, particularly suited to the short-code nature of eLoran data transmissions. In this paper, we employ the decoding method introduced in [23], which primarily results in a storage complexity of $O({2\cdot q}^{2})$ and a calculation complexity of $O({L\cdot N\cdot {log}_{2}N\cdot q}^{2})$. Given the substantial performance gains previously discussed, the complexity is considered both acceptable and feasible for implementation in short-code eLoran systems. The 128-*ary* polar code presented in this paper successfully integrates these advantages into the eLoran system. A crucial aspect of the proposed scheme is its compatibility with the existing eLoran infrastructure. By retaining 128-*ary* PPM, which is fundamental to the eLoran system’s positioning function, our approach ensures that enhancements to data communication capability do not compromise the primary navigation service.

In summary, the replacement of conventional RS codes with the proposed 128-*ary* polar code, supported by SS demodulation and SCL decoding, offers a significant improvement to the eLoran data communication system. The demonstrated gains in error performance and data efficiency are achieved while maintaining compatibility with conventional positioning functions and acceptable computational complexity.

Future work may extend the modulation set beyond balanced modulation, optimize symbol mapping through constellation shaping to enhance resilience, and further reduce the complexity of SCL decoding to improve efficiency. Conducting these studies is expected to further improve the error performance of polar codes in eLoran systems.

## Figures and Tables

**Figure 1 sensors-25-04638-f001:**
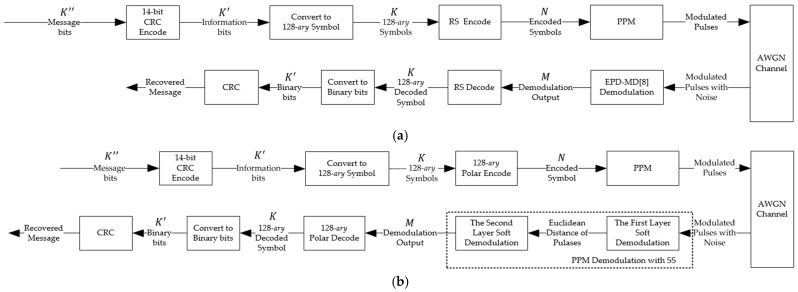
(**a**) System model of traditional eLoran data communication system with RS code and EDP-MD [8] demodulation; (**b**) system model of eLoran data communication system with 128-*ary* polar code and SS demodulation.

**Figure 2 sensors-25-04638-f002:**
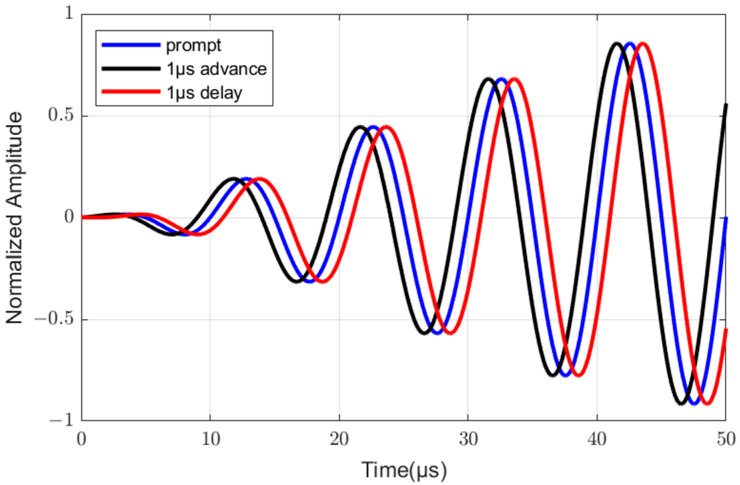
PPM signal for one pulse.

**Figure 3 sensors-25-04638-f003:**
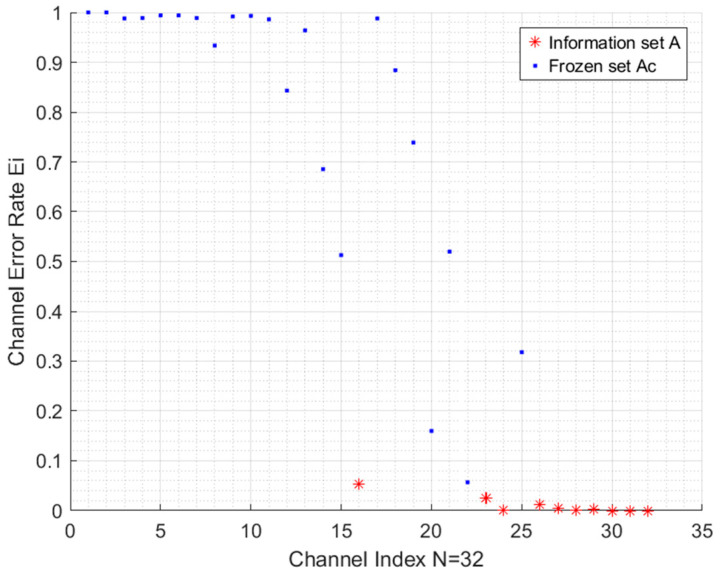
The CERs of polarized subchannels after Monte Carlo simulations with SNR=2.5.

**Figure 4 sensors-25-04638-f004:**
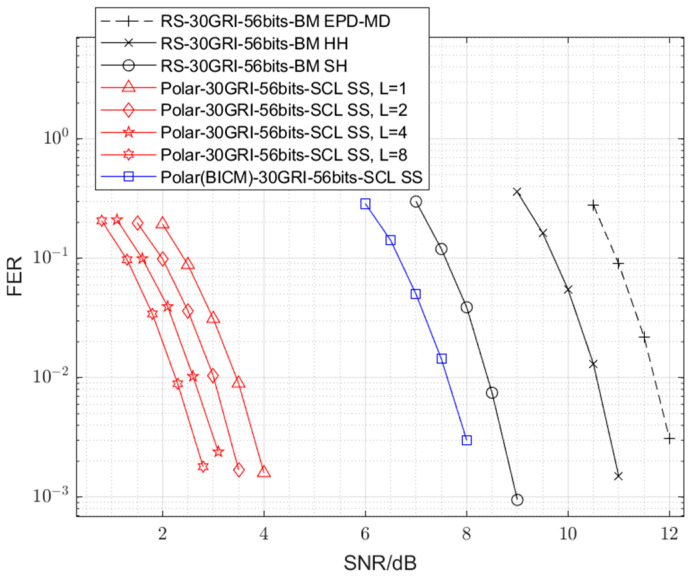
Error performance comparisons between the 128-*ary* polar code with SS demodulation, the binary polar code with BICM, and the RS code with various demodulations.

**Figure 5 sensors-25-04638-f005:**
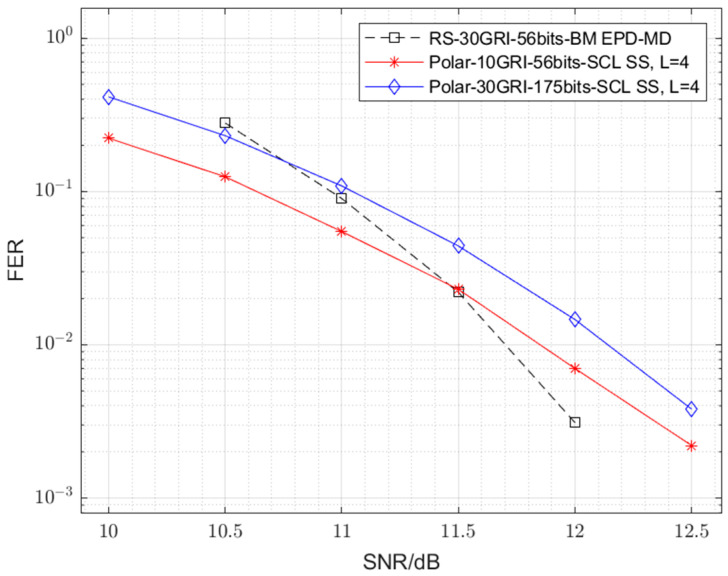
Error performance comparisons between the 128-*ary* polar code with SS demodulation (varying the GRI number and message bit number) and the RS code with EPD-MD demodulation.

**Figure 6 sensors-25-04638-f006:**
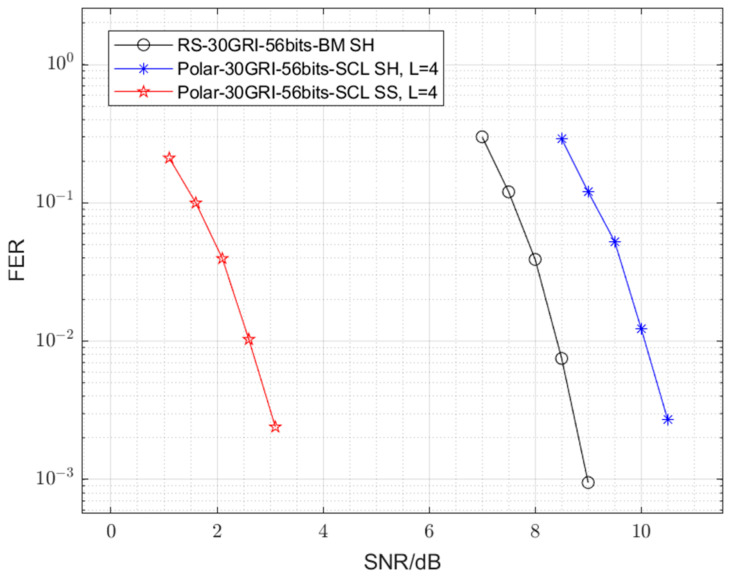
Error performance comparisons between the 128-*ary* polar code with SH demodulation and the RS code with SH and HH demodulation.

**Figure 7 sensors-25-04638-f007:**
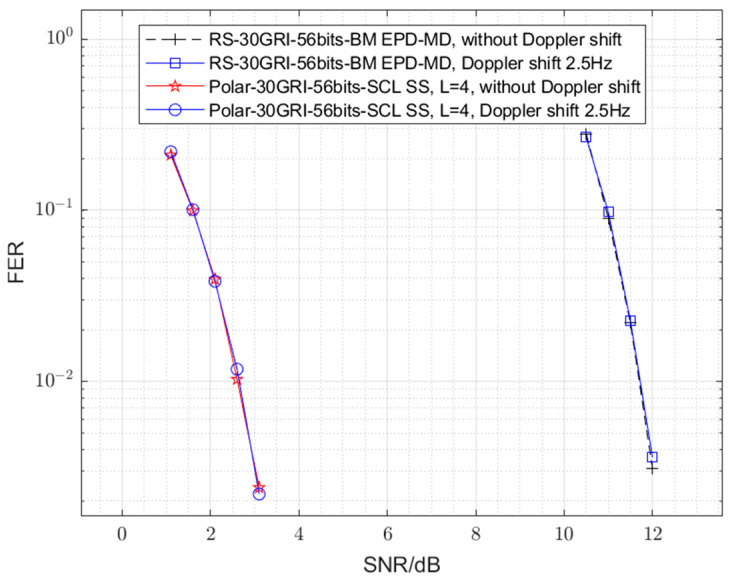
Error performance comparison between the eLoran system in the absence of Doppler shift and under a 2.5 Hz Doppler shift.

**Table 1 sensors-25-04638-t001:** Simulation parameters with the 128-*ary* polar code, the RS code, and the binary polar code.

	128-*ary* Polar Code	RS Code	Binary Polar Code (BICM)
Encoding Method	Polar Code	RS Code	Polar Code
GRI Number	30	30	30
Message Bit Number	56	56	56
CRC Bit Number	14	14	14
Coded Symbol	128-*ary*	128-*ary*	Binary
Modulation	128-*ary* PPM	128-*ary* PPM	BICM+128-*ary* PPM
Demodulation	SS/SH	SH/HH/EDP-MD [8]	SS
Decoding	SCL	BM	SCL
List size of Decoding	1/2/4/8	N/A	4
Number of Frames	$10^{6}$	$10^{6}$	$10^{6}$

**Table 2 sensors-25-04638-t002:** The error performance gain of the proposed 128-*ary* polar code with SS demodulation.

New Scheme	Error Performance Gain at 0.01 FER Level (Compared with RS and EPD-MD)
Correlation receiver	1.2 dB
Soft-decision demodulation	2.1 dB
128-*ary* polar code with SDD	6 dB
Total	9.3 dB

**Table 3 sensors-25-04638-t003:** Simulation code length, code rate, and CRC length of the 128-*ary* polar code.

	128-*ary* Polar Code-10GRI	128-*ary* Polar Code-175 bits
GRI Number	10	30
Message Bit Number	56	175

## Data Availability

The original contributions presented in this study are included in the article. Further inquiries can be directed to the corresponding authors.

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
