# Peer review of "Improving Data Communication of Enhanced Loran Systems Using 128-ary Polar Codes"

_sensors, 2025, doi:10.3390/s25154638_

Round 1

Reviewer 1 Report

Comments and Suggestions for Authors

In this paper, the authors present a novel scheme to enhance the eLoran data communication by combining the 128-ary non-binary polar code with soft-soft demodulation scheme. The gains in error performance and data efficiency have been demonstrated compared to the conventional schemes using simulation. The reviewer has the following comments to be addressed:

  • The novelty of the proposed scheme must be clarified well compared to those schemes in the literature such as in [23] and others.
  • Although the authors claim the proposed scheme does not require huge computational complexity, but the detailed complexity is not calculated and compared over the conventional schemes.
  • In the simulation part, the authors are only considering the ideal synchronization assumptions, The performance should be investigated under timing errors. Doppler shift and/or multipath.
  • Different decoder list size should be investigated and/or the selected size should be justified.
  • What is the number of framed used for the simulation.
  • The paper needs to be proofread as many grammatical errors are existed.
  • Some references are outdated; can they be replaced by newer related references?

Author Response

Thank you for your helpful comments. Please see the attachment.

Reviewer 2 Report

Comments and Suggestions for Authors

This study proposes replacing traditional RS codes with 128-ary polar codes in the enhanced Loran (eLoran) system, presenting a promising technical solution to improve the efficiency and error performance of navigation data communication. The design idea of combining polar codes with 128-ary PPM modulation has clear reference value for the communication optimization of ground-based navigation backup systems. However, the manuscript has formatting issues such as contradictory symbol definitions, incorrect formula numbering, and chaotic structural hierarchy. Moreover, references from the past five years account for less than 10%, failing to fully reflect the latest progress in the field of non-binary polar codes. It is necessary to correct the formatting, supplement recent five-year literature, and add experimental details, with a suggestion for major revision.

Author Response

(The authors gave the same response as above.)

Reviewer 3 Report

Comments and Suggestions for Authors
  1. While the paper highlights the use of 128-ary polar codes, it does not sufficiently compare this approach to other potential non-binary coding schemes (e.g., non-binary LDPC codes or turbo codes). A brief discussion or comparison with other non-binary codes would strengthen the justification for choosing polar codes.
  2. The description of the 128-ary polar code construction in Section 2.2.3 is somewhat brief. The Monte Carlo simulation method for selecting the information set A is mentioned, but key details, such as the rationale for choosing an SNR of 2.5 dB for the simulations or the computational complexity of the process, are not elaborated. Additionally, the choice of the non-binary kernel parameters (α=57\alpha = 57α=57, β=1\beta = 1β=1) is not justified
  3. The paper mentions that the proposed 128-ary polar code is not necessarily optimized (Page 5, Line 155). This statement raises questions about the potential for further improvements and whether the reported 9.3 dB gain could be increased with optimization.
  4. The comparison with the binary polar code using BICM is useful, but the paper does not discuss why the 128-ary polar code outperforms the binary polar code with BICM by such a significant margin. A deeper analysis of this performance gap would enhance the paper's technical depth
  5. The definition of SNR (Equation 20) is clear, but the paper does not specify the range of SNR values tested or the specific FER levels at which the 9.3 dB gain is measured. This makes it difficult to assess the robustness of the results across different operating conditions.
  6. The discussion of future work is limited. Given the statement that the 128-ary polar code is not fully optimized, the paper could provide more specific directions for future research.
Comments on the Quality of English Language

Must be improved

Author Response

(The authors gave the same response as above.)

Round 2

Reviewer 1 Report

Comments and Suggestions for Authors

The paper has been improved significantly and the authors addressed the comments well, but the paper needs to be proofread carefully. For instance, “The Number of Framed” should be The Number of frames, and so on.

Author Response

Comments 1: The paper has been improved significantly, and the authors addressed the comments well, but the paper needs to be proofread carefully. For instance, “The Number of Framed” should be The Number of frames, and so on.

Response 1: Thank you for pointing this out. We agree with this comment. Therefore, we have proofread the whole paper.

We hope our revisions address your concerns. Once again, we thank you for your helpful comments.

Reviewer 2 Report

Comments and Suggestions for Authors

Most issues have been revised, but there are still some remaining problems. It is suggested to make minor revisions.

  1. The numbering format of the subheadings in Sections 2.2.1 and 2.2.2 is inconsistent with that of the preceding and following headings.
  2. In Table 1, the "List size of Decoding" column for "128-ary Polar Code" includes "1/2/4/8", yet in Section 3.1, only L=4 is selected as the recommended scheme with a mere mention of "comprehensively considering complexity and error performance". Although there is a description of SCL decoding complexity in the Discussion section, the quantitative differences in complexity for different L values (i.e., L=1, 2, 4, 8) are not further elaborated. It is suggested to supplement specific quantitative data on complexity corresponding to different L values (e.g., comparative analysis of computational complexity, differences in decoding delay, etc.), so as to clearly illustrate the rationality of choosing L=4 when "comprehensively considering complexity and error performance".

Author Response

Comments 1: The numbering format of the subheadings in Sections 2.2.1 and 2.2.2 is inconsistent with that of the preceding and following headings.

Response 1: Thank you for pointing this out. We agree with this comment. Therefore, we have revised the numbering format of the subheadings.

Comments 2: In Table 1, the "List size of Decoding" column for "128-ary Polar Code" includes "1/2/4/8", yet in Section 3.1, only L=4 is selected as the recommended scheme with a mere mention of "comprehensively considering complexity and error performance". Although there is a description of SCL decoding complexity in the Discussion section, the quantitative differences in complexity for different L values (i.e., L=1, 2, 4, 8) are not further elaborated. It is suggested to supplement specific quantitative data on complexity corresponding to different L values (e.g., comparative analysis of computational complexity, differences in decoding delay, etc.), so as to clearly illustrate the rationality of choosing L=4 when "comprehensively considering complexity and error performance".

Response 2: We agree with this comment. Therefore, we have illustrated the rationality of choosing L=4 in Section 3. This choice is based on the observation from Figure 4 that increasing L from 2 to 4 yields a notable gain of approximately 0.4 dB. Since the calculation complexity is linearly related to the list size L (is discussed in Section 4), further increasing L to 8 doubles the calculation complexity while providing a negligible gain of approximately 0.2 dB. Thus, L=4 represents an efficient trade-off. These details can be found in Section 3, paragraph 3, lines 355–359 on page 10.

We hope our revisions address your concerns. Once again, we thank you for your helpful comments.